# Isolation of Bovine and Human Milk Extracellular Vesicles

**DOI:** 10.3390/biomedicines11102715

**Published:** 2023-10-06

**Authors:** Ralf Weiskirchen, Sarah K. Schröder, Sabine Weiskirchen, Eva Miriam Buhl, Bodo Melnik

**Affiliations:** 1Institute of Molecular Pathobiochemistry, Experimental Gene Therapy and Clinical Chemistry (IFMPEGKC), Rheinisch-Westfälische Technische Hochschule (RWTH) University Hospital Aachen, D-52074 Aachen, Germany; saschroeder@ukaachen.de (S.K.S.); sweiskirchen@ukaachen.de (S.W.); 2Electron Microscopy Facility, Institute of Pathology, Rheinisch-Westfälische Technische Hochschule (RWTH) Aachen University Hospital, D-52074 Aachen, Germany; ebuhl@ukaachen.de; 3Department of Dermatology, Environmental Medicine and Health Theory, University of Osnabrück, D-49076 Osnabrück, Germany; melnik@t-online.de

**Keywords:** cargo, extracellular vesicles, intercellular signaling, NGS, electron microscopy, metabolic imprinting, immune regulation, epigenetic, milk microvesicles

## Abstract

Extracellular vesicles such as exosomes are small-sized, bilayered extracellular biovesicles generated by almost every cell and released into the surrounding body fluids upon the fusion of multivesicular bodies and the plasma membrane. Based on their origin, they are enriched with a variety of biologically active components including proteins, lipids, nucleic acids, cellular metabolites, and many other constituents. They can either attach or fuse with the membrane of a target cell, or alternatively be taking up via endocytosis by a recipient cell. In particular, milk exosomes have been recently shown to be a fundamental factor supporting infant growth, health, and development. In addition, exosomes derived from different cell types have been shown to possess regenerative, immunomodulatory, and anti-inflammatory properties, suggesting that they are a potential therapeutic tool in modulating the pathogenesis of diverse diseases. Therefore, efficient protocols for the isolation of milk exosomes in a high quantity and purity are the basis for establishing clinical applications. Here, we present an easy-to-follow protocol for exosome isolation from bovine and human milk. Electron microscopic analysis and nanoparticle tracking analysis reveal that the protocols allow the isolation of highly enriched fractions of exosomes. The purified exosomes express the typical exosomal protein markers, CD81 and ALIX.

## 1. Introduction

Milk is a highly complex nutrient developed in mammalian evolution to promote postnatal growth [1,2]. It has a high nutritional value enriched with proteins and contains calcium and vitamin D that are of pronounced importance for bone formation and mineralization [3]. Furthermore, it has been recognized that milk functions as a postnatal epigenetic imprinting system that impacts metabolic programming and protects against diseases in later life (e.g., heart disease, high blood pressure, and diabetes) [4,5].

In 2007, it was shown for the first time that human breast milk contains large quantities of extracellular vesicles (exosomes and microvesicles), which are supposed to be important for the development of an infant’s immune system [6]. Previous studies have shown that bovine milk contains about 10^12^–10^14^ exosomes/mL, while the content in human milk is 50–270% lower [7,8]. Other studies found that the exosome concentration in human milk is the highest in colostrum (the first stage of breast milk that lasts for some days after childbirth), while the concentration is significantly lower in transitional and mature milk (distinct later stages of breast milk production) [9]. Recent findings further suggested that the cargo of exosomes included in milk products has beneficial effects on metabolic interorgan cross-talk [10].

In general, exosomes are small phospholipid bilayered extracellular spherical vesicles, which are secreted by almost all cells. These cellular nanospheres are capable of transferring cell-specific constituents (i.e., the cargo) from a source cell to a recipient cell. Therefore, exosomes are commonly accepted as ubiquitous mediators of intercellular communication. They are approximately 40–150 nm in diameter and are formed by a multivesicular endosomal route [11]. Exosomes have an important function in both physiological and pathological processes. On the one side, they induce immunological responses, promote angiogenesis, and improve neurogenesis [12,13,14,15], while on the other hand, they drive the proliferation and migration of cancer cells, contribute to blood-brain barrier damage after a cerebral ischemia, and contribute to pain formation in cancer [15,16,17].

In addition, based on their low immunogenicity, mechanical properties, and prolonged blood circulation, nowadays, exosomes, in general, are recognized as potential cutting-edge drug carrier and delivery systems [18]. In this regard, the methods for the production of engineered exosomes and drug loading are already well established, and these respective systems are already applied as safe drug carriers in various therapies [19].

Exosomes from different sources can be isolated using a number of different techniques, including differential centrifugation, density gradient centrifugation, ultrafiltration, size-based isolation techniques, immunoaffinity capture-based methods, precipitation protocols, and microfluidics-based isolation techniques [20]. All these methods have their advantages and disadvantages. Although differential ultracentrifugation is currently considered as the “gold standard” for exosome isolation, exosome preparations produced using this method are often contaminated by large quantities of proteins and lipoproteins [20]. Nevertheless, ultracentrifugation-based techniques have several advantages. They have a large sample capacity, yield large amounts of exosomes, and the costs of respective isolation are rather low when the equipment (e.g., centrifuges and ultracentrifuges) is present. However, when using milk as a starting material, the application of standard ultracentrifugation protocols is hindered by the fact that exosomes generated via this method are often contaminated by proteinaceous casein-containing aggregates that routinely co-sediment with the exosome particles during the purification process [21,22].

There are many protocols available for the isolation of milk exosomes from bovine and human milk. Most protocols rely on differential ultracentrifugation, size chromatography, serial filtration, sucrose gradient centrifugation, or commercially available polymer-based exosome precipitation kits such as ExoQuick^TM^ [21,23,24,25]. Most recently, a scalable protocol for the high-quality exosome purification of bovine milk that uses electrophoretic oscillation-assisted tangential flow filtration with the antifouling of micro-ultrafiltration membrane filters was introduced [26]. Another protocol combines centrifugation, filtration, and a polyethylene glycol precipitation step to isolate exosomes from human breast milk [27]. In addition, the combination of consecutive ultracentrifugation, gel filtration, and affinity chromatography on anti-CD9- and anti-CD63-sepharose was applied to isolate ultrapure exosomes from horse milk [28]. Finally, bovine milk exosomes were successfully purified using a high-resolution size-guided turbidimetry-enabled particle purification liquid chromatography (PPLC) method that combines gradient size exclusion column, fraction collection, and online UV-Vis/turbidimetry absorbance measurements calculations [29]. However, all these methods are not cost-effective and require special equipment or reagents, thus preventing their applicability in isolating large quantities of exosomes in a standard laboratory.

In this article, we provide a simple protocol for the isolation of exosomes from bovine and human milk. This protocol is based on a sequence of standard and ultracentrifugation steps, filtration through 0.2 µm filters, and the removal of contaminating casein aggregates via a treatment with ethylenediaminetetraacetic acid (EDTA). Electron microscopic analysis and nanoparticle tracking analysis demonstrate that the protocol allows the isolation of highly enriched exosome fractions. Western blot analysis revealed that the purified particles express typical exosomal protein markers such as CD81 and ALIX.

## 2. Experimental Design

Like previous protocols, the protocol for the isolation of milk exosomes is principally based on centrifugation and filtration steps. In addition, a treatment with EDTA was used to remove subcritical amounts of Ca^2+^ from casein micelles, thereby dissociating caseins and releasing soluble casein and minerals from casein micelles via chelating calcium [30]. The final protocol schematically outlined in Figure 1 is suitable to purify large quantities of casein-free milk exosomes in about five hours.

## 3. Materials and Equipment

### 3.1. Sources of Milk

#### 3.1.1. Bovine Milk

Fresh bovine raw milk can be obtained from any dairy raw milk vending machine and direct marketers. As starting material for our study, we used milk from a traditional dairy cattle farm (Familie Joachim Stauten, Willich, Germany; https://www.landservice.de/bauernhoefe/Familie-Stauten/natur-care; accessed on 2 September 2023). The milk on this farm is largely taken from healthy lactating dairy cows of the type Holstein Schwarzbunt. The milk was drawn and stored in glass bottles at 4 °C one day before exosome preparation (Figure 2).

#### 3.1.2. Human Breast Milk

Our study used 15 mL-30 mL milk samples from healthy mothers with sufficient milk production enrolled in the Department of Obstetrics and Gynecology, RWTH University Hospital Aachen, Aachen, Germany. The samples were stored in 130 mL plastic sampling flasks and transferred to the registered RWTH centralized Biomaterial Bank of the Medical Faculty in Aachen. After they were de-identified, the samples were acquired by us on the same sampling day and stored at 4 °C (Figure 3). The isolation of exosomes from these samples was either initiated on the same day or the following day.

Note: (i) Following this strategy of sample recruitment, potential ethical considerations are fully accounted for on the basis of the existing positive ethics vote of the Faculty of Medicine of the RWTH University Aachen and the consent of the donors that provided the samples. (ii) Actually, a common legislative framework of usage donor human milk for scientific purposes is lacking, and national legislative frameworks might differ widely. Therefore, we recommend informing yourself in advance before using human milk as a starting material. (iii) Please note that the composition of human breast milk and the exosomes derived thereof might change in different phases. The color of the milk transition from colostrum (days <1–4) to transitional milk (days 5–20) and mature milk (days 20+) slowly changes from yellow to white.

### 3.2. Buffers and Solutions

#### 3.2.1. 200 mM HEPES Buffer (pH 7.4)

A total of 47.66 g 4-(2-hydroxyethyl)-1-piperazineethanesulfonic acid (HEPES, CAS no.: 7365-45-9; molecular weight: 238.31 g/mol) was dissolved in 800 mL distilled autoclaved water, and the solution adjusted to a pH of 7.4 by adding 10N NaOH solution. The solution was then filled up to 1 L with autoclaved water and filtered through a 0.2 µm syringe pore filter for sterilization and kept at 4 °C away from visible light. HEPES from Merck (Sigma-Aldrich, Taufkirchen, Germany; #PHG0001) was used in our study. Note: (i) The exposure of HEPES-containing solutions to visible light leads to the generation of cytotoxic substances [31]. Although HEPES has a melting point of >234–238 °C (453–457 K), and should therefore resist standard autoclaving, many protocols advise against sterilizing HEPES solutions via standard autoclaving (121 °C, 20 min). (ii) Some companies offer ready-to-use cell culture-grade HEPES buffer solutions with a low endotoxin content. These buffers should be preferably used if the prepared exosomes should be used in cell culture experiments.

#### 3.2.2. 50 mM HEPES Buffer (pH 7.4)

A 50 mM HEPES buffer, pH 7.4: 100 mL solution was prepared by mixing 25 mL 200 mM HEPES (pH 7.4) with 75 mL sterile water.

#### 3.2.3. 500 mM EDTA Solution (pH 7.5)

To prepare 1 L of 500 mM ethylenediaminetetraacetic acid solution (EDTA, CAS no.: 60-00-4; molecular weight: 292.24 g/mol), 186.1 g EDTA × 2 H_2_O was added to 800 mL distilled water, and the solution was stirred vigorously with a magnetic stirrer. The pH was adjusted to 7.5 with 10N NaOH and the solution filled to a volume of 1 L and sterilized via autoclaving (121 °C, 20 min). Note: The EDTA will turn into a solution without adding NaOH.

#### 3.2.4. Protein Lysis Buffer

Protein lysis buffer consisted of 20 mM Tris-HCl (pH 7.2), 150 mM NaCl, 2% (*w*/*v*) NP-40, 0.1% (*w*/*v*) SDS, and 0.5% (*w*/*v*) sodium deoxycholate and was stored at −80 °C. Immediately before use, the lysis buffer was supplemented with a mixture of Complete^TM^ proteinase inhibitor (Merck, Darmstadt, Germany; #11697498001) and phosphatase inhibitor cocktail II (Sigma-Aldrich; #P5726-1ML).

### 3.3. Consumables

#### 3.3.1. Sterile Plastic Tubes with Screw Caps (50 mL)

Different brands of 50 mL sterile transparent pyrogen-free polypropylene centrifuge tubes with a screw cap, a conical base, and writing space and graduation can be used. Either caps from Sarstedt (Nürnbrecht, Germany; #62547254; 114 × 28 mm), or alternatively, from Corning Life Sciences/Falcon (Kaiserslautern, Germany; #352070; 30 × 115 mm) were used. Note: These tubes allow centrifugation at a maximum of 20,000× *g*.

#### 3.3.2. Sterile Ultracentrifuge Tubes (14 mL)

Appropriate tubes designed for ultracentrifugation are necessary. In the presented study, we exclusively used open-top polypropylene round bottom tubes (14 × 95 mm) from Beckman Coulter (Krefeld, Germany; #331374) that are compatible with the swinging bucket ultracentrifuge rotor SW 40 Ti from Beckman Coulter. The tubes can be autoclaved prior use. Note: For the cleaning and re-use of the respective tubes, refer to the instructions of the supplier. In this study, the tubes were discarded after first use.

#### 3.3.3. Plastic Syringes (50 mL with Luer Lock Fitting)

Sterile, transparent 50 mL polypropylene perfusor luer lock syringes are available from many suppliers. In this study, syringes from B. Braun Petzold (Melsungen, Germany; #8728844F-06) were used because the pistons of these syringes have excellent sliding properties and a double sealing ring, which prevents leakage and ensures a minimum residual volume. Their male luer lock outlet fitting allowed us to connect them to suitable filters with the luer inlet lock fitting.

#### 3.3.4. Syringe Filters (with Pore Size 0.20 µm)

Standard sterile syringe filters with polypropylene housing, a pore size of 0.20 µm, and a low hold-up volume are required for the removal of remaining non-exosomal particles after the first ultracentrifuge run. These can be obtained from any supplier. In this study, surfactant-free, cellulose acetate membrane filters from Corning Life Sciences (#431219) were used. These filters have a female luer inlet lock fitting.

#### 3.3.5. Serological Pipets (10 mL and 25 mL)

These pipets are available from many suppliers. Sterile and pyrogen-free serological polystyrene pipets were obtained from Corning Life Sciences (10 mL: #4488, 25 mL: 4489).

#### 3.3.6. Lint-Free Cloths

Standard soft, absorbent, lint-free cloths can be obtained from any supplier such as from Kimberly-Clark GmbH (Mainz, Germany; #33670-04), or Papernet (Vienna, Austria; #415938) or from Lucart professional (Neusäß, Germany; #841030).

#### 3.3.7. Sterile Scalpels

Ready-to-use disposable sterile surgical scalpels with stainless steel blades are offered by various providers. Large curved blades (#10, #21, and #22) are the most suitable. Suitable disposal scalpels are from Feather Safety Razor Co., Ltd. (Ooyodominami, Osaka, Japan; #FB.21), which were used in this study. Note: When using the scalpels for cutting the bottom of the ultracentrifuge tubes (in Section 4.4), please follow the safety instructions: (i) Safety glasses may be worn for protection when cutting. (ii) Small blades such as #11 are not suitable because they have been designed for limited lateral pressure and might break during the intended use in this protocol (Figure 4).

### 3.4. Instrumentation

#### 3.4.1. Standard Refrigerated Laboratory Table Centrifuge

In our study, a Labofuge 400R/Functional Line from Heraeus Instruments (Hanau, Germany) equipped with a swing-out rotor (#8179), matching round buckets (#8172), and suitable bucket adapters allowing the centrifugation of conical 50 mL tubes was used.

#### 3.4.2. Refrigerated Floor Centrifuge with Appropriate Rotor and Bottles

A refrigerated floor-standing centrifuge for performing high-speed separations (i.e., 10,000× *g*) is required. The instrument should be able to maintain the rotor temperature to a specified temperature (i.e., 4 °C) and allow the separation of < 200 mL in one run. A Beckman Avanti J-25 high-speed centrifuge (Beckman Coulter) equipped with a fixed JLA 10.500 angle rotor (#369681) was used. This configuration allows centrifugation with a maximal speed of 10,000 rounds per minute (~18,600× *g*). The samples were centrifuged in 250 mL polypropylene bottles with screw-on cap (#356011) and suitable polyterephthalate bottle adapters (#362750) for the chosen rotor.

#### 3.4.3. Ultracentrifuge with Appropriate Rotor

An ultracentrifuge equipped with a swinging bucket rotor that allows the spinning of samples at exceptionally high speeds and capable of generating high accelerations (<100,000× *g*) is required. A Beckman Optima^TM^ L-70K ultracentrifuge (Beckman Coulter) equipped with a swinging-bucket SW 40 Ti rotor package (#331301) was used. Note: The maximum speed of this rotor is 40,000 rounds per minute, which corresponds to a relative centrifugal field (RCF) of 120,000× *g* at r_min_ (66.7 mm) and 285,000× *g* at r_max_ (158.8 mm).

#### 3.4.4. Variable Micropipettor and Suitable Sterile Tips

Variable micropipettors that allow the pipetting and transferring of small amounts (<1 mL) of liquids are necessary. The brand of micropipettor used is irrelevant, but the pipettors used should allow the user to pipet volumes of 100 µL and 500 µL. Research plus pipettes from Eppendorf (10–100 µL pipettor: #3123000047 and 100 µL–1 mL pipettor: #3123000063) were used in the presented study.

### 3.5. Antibodies

#### 3.5.1. CD81 Antibody

A mouse monoclonal antibody against amino acids 90–210 of CD81 of human origin (CD81 (B-11), Santa Cruz Biotech., Santa Cruz, CA, USA; #sc-166029) that is cross-reactive with CD81 from mice, rats, and humans was studied using Western blot analysis. Since bovine and human CD81 share 94% sequence identity (human CD81 accession no.: NP_004347.1; bovine CD81 accession no.: NP_001030271), this antibody also reacts with bovine CD81. Suitable antibodies are also available from many other suppliers. Note: CD81 also known as 26 kDa cell surface protein, Tetraspanin-28 (TSPAN28), or a target of antiproliferative antibody 1 (TAPA-1), which is a cell surface glycoprotein of the transmembrane 4 superfamily that is known to complex with integrins [32]. CD81 is conventionally used as a marker for exosomes [33].

#### 3.5.2. ALIX Antibody

In addition, a mouse monoclonal antibody against full-length ALIX of human origin (ALIX (1A12), Santa Cruz; #sc-53540) that is cross-reactive with ALIX from mice, rats, and humans was used. Since bovine and human ALIX share 95% sequence identity (human ALIX access. no.: AAH20066; bovine ALIX access. no.: XP_005222647), this antibody also detects bovine ALIX. Suitable antibodies are also available from many other suppliers. Note: ALG2-interacting protein (ALIX) that is also commonly known as programmed cell death 6-interacting protein (PDCD6IP) is essential for clathrin-independent endocytosis and signaling, membrane deformation, and fission both in endosomes and at the plasma surface [34]. ALIX is one of the best-established exosome markers with essential functions in regulating exosomal secretion [35].

#### 3.5.3. β-Actin

For testing the protein integrity in exosome fractions, the blots were routinely probed with an antibody directed against β-actin (β-actin (AC-15), Sigma-Aldrich, Taufkirchen, Germany; #A5441).

Note: In our study, exosome fractions prepared from both human and bovine milk are positive for β-actin. However, it should be noted that the occurrence of β-actin in exosomes is actually a matter of debate. While some investigators argue that exosomes are negative for the expression of β-actin [36,37,38], others found β-actin and other actin-associated proteins in exosomes from various sources [39,40,41,42,43,44]. In contrast to exosomes prepared from milk, exosomes derived from cultured A549 and PC-3 cells were found to be negative for β-actin expression [44]. It might be possible that elements of the actin cytoskeleton may be trapped in the exosomes of particular sources as a result of the cytoskeleton’s involvement in the exosome formation process [43].

## 4. Detailed Procedure

The isolation of exosomes from bovine and human milk was carried out in several standard or ultracentrifuge centrifugation steps, each with an increasing force. The presented protocol used either 100 mL bovine milk or a volume of 15 mL human milk as the starting material. However, the protocol is scalable to all other volumes. All buffers should be ice-cooled, and each centrifugation step should be performed in refrigerated centrifuges at 4 °C. The consumables and equipment given are those used in our laboratory. These can be replaced with equivalents from other suppliers and companies. It should be once noted that the composition of milk and exosomes derived thereof might change in different phases (see also the Note in Section 3.1.2).

### 4.1. Removal of Fat Globules, Cream, Cellular Debris, and Somatic Cells

Transfer the milk into two (100 mL bovine) tubes or one (15 mL human) 50 mL tube.Centrifuge the tubes in a standard refrigerated table centrifuge for 30 min at 3000× *g*. Note: This step allows the removal of fat globules and cream that will float after centrifugation on the supernatant.Remove the fat globule/cream fraction that is on the top of the supernatant (Figure 5) with a sterile cube.

**Figure 5 biomedicines-11-02715-f005:**
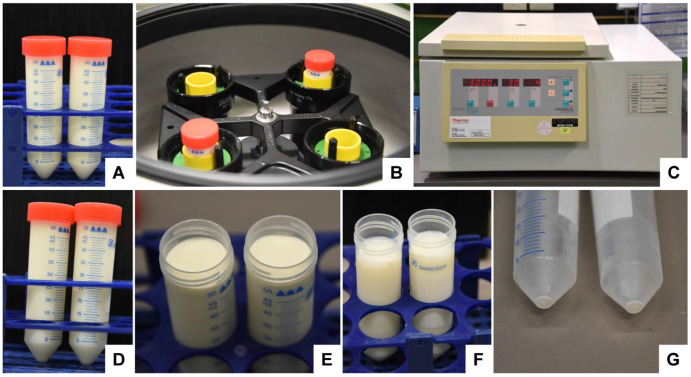
The removal of fat globules and cream from bovine milk via standard centrifugation. (**A**) Bovine raw milk is (**B**,**C**) centrifuged in a standard laboratory centrifuge for 30 min at 3000× *g*. (**D**–**F**) After centrifugation, a thick fat layer of cream floats on the surface, which can be easily removed from the surface. (**G**) After the removal of the skimmed milk, a small pellet is visible at the bottom of the tubes.

After the removal of the fat globule/cream fraction, transfer the defatted milk solution into suitable centrifuge tubes.Centrifuge the tubes in a floor centrifuge at 10,000× *g* for 30 min. Note: (i) This step allows the removal of cellular debris and somatic cells. (ii) In our laboratory, we transferred the solution to 250 mL polypropylene bottles (2 for bovine samples, and 1 for human sample) and centrifuged the bottles in a Beckman Avanti J-25 high-speed centrifuge (Beckman Coulter, Krefeld, Germany) equipped with a fixed JLA 10.500 angle rotor (10,000× *g* = 7334 RCF) (Figure 6).

**Figure 6 biomedicines-11-02715-f006:**
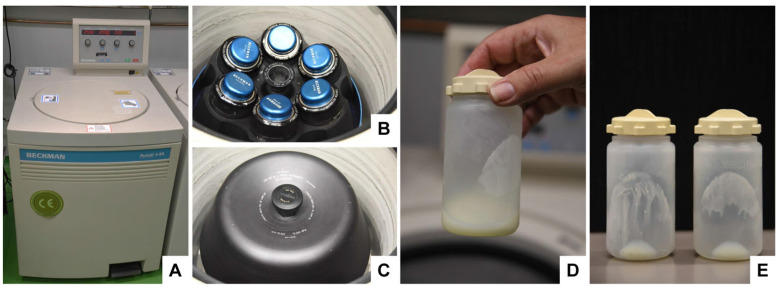
The removal of cellular debris and somatic cells from skimmed bovine milk. (**A**–**C**) Centrifugation in this step is performed in a floor centrifuge equipped with a suitable rotor and buckets. (**D**,**E**) Well-recognizable pellets are formed, and remaining cream is attached to the wall of the centrifugation bottles, which are very visible after the pouring of the supernatant.

Transfer the supernatants into sterile 50 mL centrifuge bottles (bovine: 3 tubes; human: 1 tube) and discard the pellets.Estimate the volume in each tube.

### 4.2. Solubilisation of Casein-Containing Protein Aggregates

Add 500 mM EDTA (pH 7.5) to each tube to reach a final EDTA concentration of 125 mM (i.e., add 3.33 mL per 10 mL defatted milk solution).Shake the mixture gently for 30 min at 4 °C. Note: In this step, the removal of subcritical amounts of divalent calcium ions from casein micelles in EDTA releases soluble casein [30]. The white cloudiness of the milk will disappear, and the milk solution will become translucid (Figure 7).

**Figure 7 biomedicines-11-02715-f007:**
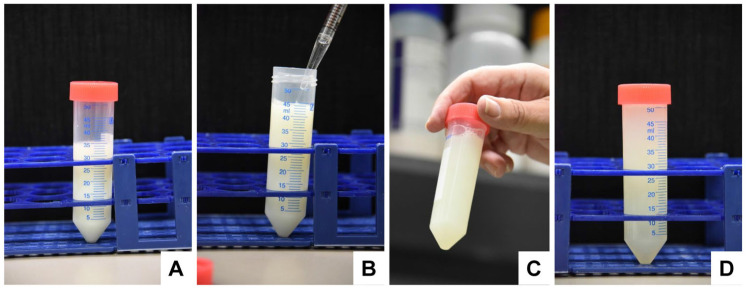
The clearance of milk from casein-containing aggregates. (**A**–**D**) The addition of EDTA to the defatted milk releases soluble casein, and the cloudiness of the milk solution disappears.

Transfer the solution into 250 mL centrifuge bottles (bovine: 2; human: 1) and centrifuge the tubes in a floor centrifuge at 10,000× *g* for 30 min as described above.

### 4.3. Precipitation of Exosomes via Ultracentrifugation

Transfer the resulting translucid supernatant to sterile ultracentrifuge tubes (bovine: 6; human: 1) and discard the pellet(s). Note: (i) Please read the safety instructions and refer to the instrument instructions before performing any ultracentrifuge run. (ii) In most devices, it is necessary to hook all the buckets, loaded or empty, to the rotor when operating. (iii) Fill the tubes almost to the top edge to prevent the collision of the tubes during the centrifuge run.Prepare the tare tubes. Note: (i) In the case of the human sample, prepare a tare tube. (ii) In our laboratory, we use a Beckman Optima^TM^ L-70K ultracentrifuge (Beckman Coulter, Krefeld, Germany) equipped with an SW 40 Ti rotor (round per minute (rpm) = 29,500; average relative centrifugal force (RCF_avg_) = 109,895; maximal relative centrifugal force (RCF_max_) = 154,779; k-factor: 252.5) to pellet the exosomes. (iii) If you work with another ultracentrifuge or rotor, please refer to the handbook of your instrumentation for g-force to rpm conversion, or alternatively calculate the necessary rpm for the respective ultracentrifuge according to the following equation:

rpm = √[RCF/(*r* × 1.118)] × 10^5^


(in which *r* is the rotational radium (in cm).

Centrifuge the tubes at 100,000× *g* for 70 min.After centrifugation, remove the supernatants. Note: The solution should look transparent at this stage (Figure 8A). (ii) It might be possible that on the top of the supernatant, a thin layer of fat is visible (Figure 8B). Take care that this film is properly removed from the supernatant.Remove the remaining fluid within tubes by placing the liquid-free pellet-containing tubes onto a sterile, absorbent, lint-free cloth paper (Figure 8C).

**Figure 8 biomedicines-11-02715-f008:**
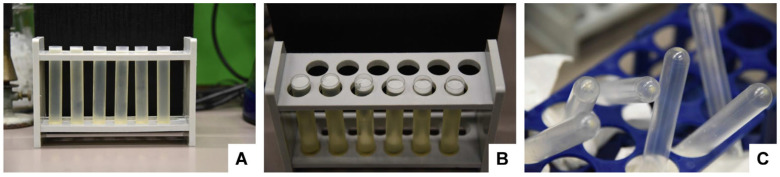
The enrichment of exosomes via ultracentrifugation. (**A**) After ultracentrifugation, the supernatant should look transparent. (**B**) Foamy layers might float on the supernatant. (**C**) After the removal of the supernatant, clear pellets enriched in exosomes should be obtained. The removal of remaining supernatant is achieved by transferring the liquid-free pellet-containing tubes onto sterile, absorbent, lint-free cloth papers.

Resuspend the clear exosome-enriched pellet carefully into 50 mL (bovine) or 15 mL (human) 200 mM HEPES (pH 7.4). Note: The pellets are gelatinous and not easy to dissolve. Avoid a harsh treatment and minimize the creation of foam during this step. This will damage the exosomes. Be sure that the pellets are well solved before performing the next step.

### 4.4. Filtration and Washing of Exosomes

Remove the plunger stopper from the housing of a sterile 50 mL syringe.Fix the luer lock of the syringe housing onto a sterile 0.20 µm filter.Transfer the exosome-enriched solution into the syringe housing.Carefully insert the plunger of the syringe and filter the solution.Collect the flow in a sterile 50 mL tube (Figure 9).

**Figure 9 biomedicines-11-02715-f009:**
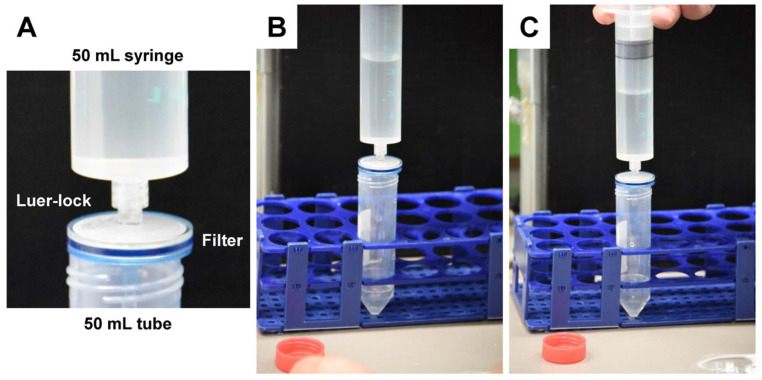
The sterile filtration of the exosome-enriched fraction. After the first ultracentrifugation step, the exosomes are dissolved in 200 mM HEPES buffer. (**A**–**C**) The solution is then pushed through a 0.20 µm pore filter using mechanical filtrations using a 50 mL syringe that is luer-locked to the filter. The procedure is repeated once by exchanging both the syringe and the filter.

Repeat the filtration step. Use a new sterile 50 mL syringe and a new 0.20 µm filter.Transfer the resulting supernatant to 3 (human) sterile ultracentrifuge tubes or 1 (human) tube and repeat the ultracentrifugation step in Section 4.3. After centrifugation, the supernatants should look transparent (Figure 10A).Carefully remove the supernatant and remove remaining fluid within tubes by transferring the liquid-free pellet-containing tubes onto a sterile, absorbent, lint-free cloth paper (Figure 10B).

**Figure 10 biomedicines-11-02715-f010:**
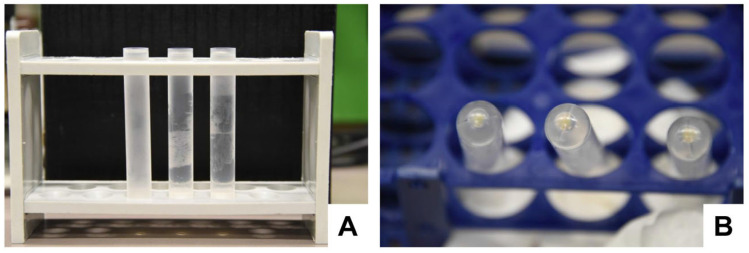
The washing of the exosome pellet in a second ultracentrifugation step. (**A**,**B**) After the second round of ultracentrifugation, the resulting supernatants should look clear, and the pellets should look gelatinous and light yellow.

Cut the bottoms of the tubes with a sterile sharp, thin-bladed scalpel (Figure 11). Note: This step has a high risk of injury. Be careful and make sure that the blade does not break off while cutting. Safety glasses may be worn for protection when cutting.

**Figure 11 biomedicines-11-02715-f011:**
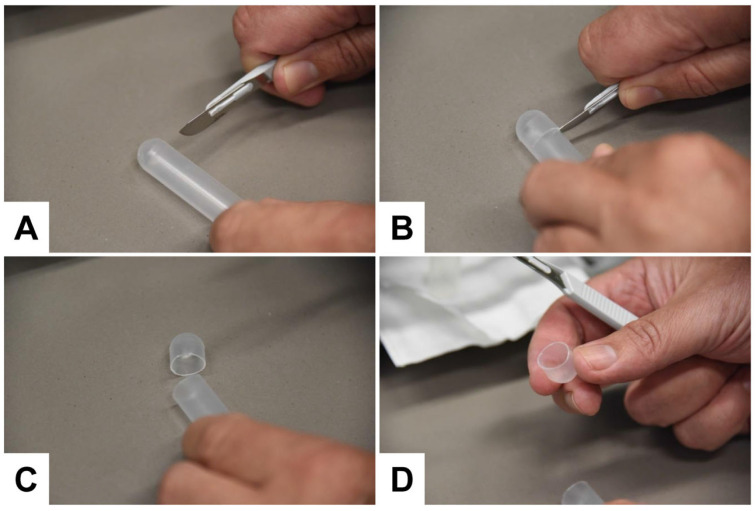
Cutting off the bottom of the centrifuge tubes. (**A**) The tube is placed on a flat surface, and the (**B**,**C**) bottom is removed by smoothly cutting it with a sterile sharp scalpel. (**D**) After the removal of the bottom, the pellet is easily accessible for pipetting out the exosome pellet.

Resuspend the pellets of all gradients and combine them in a final volume of 500 µL (bovine) or 100 µL (human) 200 mM HEPES (pH 7.4) buffer. If the exosomes are analyzed via Western blot analysis, resuspend the pellets in protein lysis buffer instead of 200 mM HEPES (pH 7.4).

### 4.5. Characterisation of Exosome Preparations

The protocol described here allows the isolation of highly pure fractions of exosomes at high yield. To document the purity and integrity of the exosomes generated with this protocol, we routinely performed Western blot analysis and transmission electron microscopy.

#### 4.5.1. Analysis of Exosomal Marker Proteins via Western Blot Analysis

The identification of typical exosomal markers is easily possible by testing for the expression of respective proteins (i.e., CD81 and ALIX) via Western blot analysis [35,45,46]. The predicted molecular weights of the respective proteins are between ~22 kDa (ALIX) and ~25.8 kDa (CD81). In our laboratory, we prefer to use preformatted 4–12% Bis-Tris gels (Invitrogen, Darmstadt, Germany) and a 3-*N*-(morpholino)propanesulfonic acid (MOPS) running buffer using other protocols published before [47]. The detection of human and bovine CD81 and ALIX proteins is easily achievable with commercially available antibodies (Figure 12).

#### 4.5.2. Determination of Size and Concentration of Exosomes via Nanoparticle Tracking Analysis

The size and concentration of exosomes in a solution can be determined via nanoparticle tracking analysis (NTA) [48,49,50]. Using this method, the sample to be analyzed is illuminated with a laser beam, and the suspended particles are visualized in real time by light scattering using a light microscope [51]. A video showing the Brownian motion of respective particles is taken during a defined time course, and the NTA software is used to calculate the hydrodynamic size via a modified Stokes-Einstein equation [50,51]. In addition, the measured real-time events per defined time interval provide a rough estimate of the concentration of particles within the solution. In our laboratory, a NanoSight NS300 instrument (Malvern Panalytical Ltd., Malvern, Worcestershire, UK) was used to measure the particle size and concentration in exosome solutions (Figure 13 and Appendix A). This device allows the measurement of 10^6^–10^9^ particles per mL in the 10 nm–2000 nm diameter range.

Note: In our analysis, the mean size of bovine exosome diameters as assessed via NTA analysis was found to be 267.2 nm ± 98.1 nm (cf. Figure 13). In contrast, the reported sizes that are characteristic of exosomes purified from other sources are in the range of 40–150 nm [11]. In this context, it should be noted that the diameter of bovine milk exosomes strongly depends on the course of lactation. In particular, the exosomes from mature milk that were used in our study as a starting material have significantly larger mean sizes than those obtained from colostrum samples [52]. As such, the terms exosomes and extracellular microvesicles are a bit vague. Consequently, researchers now encouraged the use of the term extracellular vesicles as a consensus generic term for all lipid bilayer-delimited particles released from a cell [53,54]. However, several researchers still prefer to distinguish exosome and non-exosome extracellular vesicles that can be separated as subsets at different ultracentrifugation speeds. Although commonly used extracellular vesicle markers are not differentially enriched in these different subsets, several proteins were identified that are enriched in one or the other fractions [55].

#### 4.5.3. Imaging of Purified Exosomes via Transmission Electron Microscopy

The sample preparation, negative staining, and imaging of exosomes via transmission electron microscopy (TEM) were performed as previously described in a detailed step-by-step protocol, omitting the paraformaldehyde fixation step when the samples were analyzed on the same day [56]. For our analysis, the final samples were analyzed using a transmission electron microscope (Hitachi HT7800; Tokyo, Japan) operating at an acceleration voltage of 100 KV (Figure 14A,B). Representative images of exosomes were taken over a magnification range from 20,000× to 150,000× (Figure 14C–F).

It should be noted that omitting the EDTA step (see Section 4.2) will result in exosome preparations that are contaminated with casein aggregates that can be detected using TEM (Figure 15).

## 5. Conclusions

The protocol presented in this article allows the purification of extracellular vesicles, such as exosomes from bovine and human milk, via ultracentrifugation. Although the protocol presented here was mainly established by using mature milk from dairy cows as the starting material, we demonstrate that the protocol is also suitable to isolate exosomes from human milk. However, it should be kept in mind that the concentration and size of exosomes in both bovine and human milk strongly depend of the stage of milk production. It is now necessary to establish more accurate data on the size and concentration of exosomes that can be isolated from colostrum, transitional, and mature milk. The addition of EDTA is a central step in the described purification protocol. This chelating agent sequesters divalent calcium ions and breaks down contaminating casein aggregates that are partially co-precipitated during the enrichment of exosomes. Although we have systematically analyzed how the EDTA step affects the overall yield obtained, this step is of fundamental importance to remove casein aggregates. Using TME and NTA demonstrated that the purified exosomes have a mean size of 267.2 nm ± 98.1 nm. In our study, this protocol allowed us to isolate about 5 × 10^10^ exosomes from 1 mL bovine raw milk. The purified exosomes isolated using this protocol express typical exosomal marker proteins (CD81 and ALIX). Based on the simplicity of the protocol, we hope that it will assist biomedical researchers in unravelling milk exosomes’ functionalities.

## Figures and Tables

**Figure 1 biomedicines-11-02715-f001:**
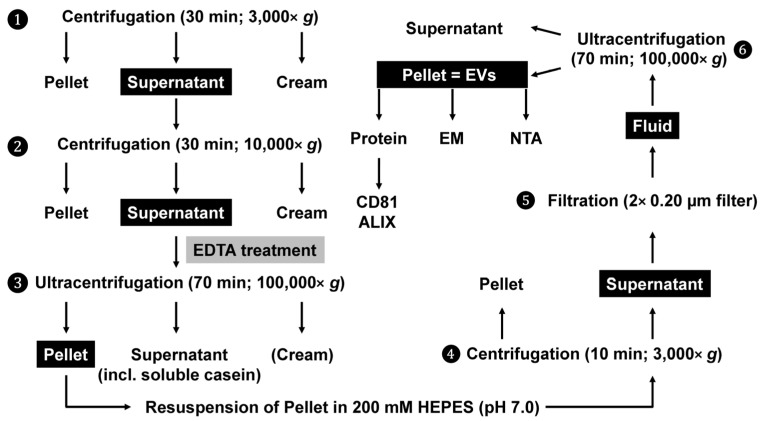
The isolation of exosomes from human and bovine milk. The milk is pretreated with initial standard centrifugation to remove fat, cellular debris, and somatic cells. Thereafter, the solution is treated with EDTA to clear casein-containing aggregates. This step removes Ca^2+^ from casein micelles, thereby destroying the micellar casein framework. Subsequently, soluble casein is removed, and exosomes are enriched via ultracentrifugation. The remaining contaminating apoptotic bodies and cell debris are removed via two rounds of filtration, and the exosomes are finally enriched via a second ultracentrifugation step. The abbreviations used are EDTA, ethylenediaminetetraacetic acid; EM, electron microscopy; EVs, extracellular vesicles/exosomes; HEPES, 4-(2-hydroxyethyl)-1-piperazineethanesulfonic acid; NTA, nanoparticle tracking analysis.

**Figure 2 biomedicines-11-02715-f002:**
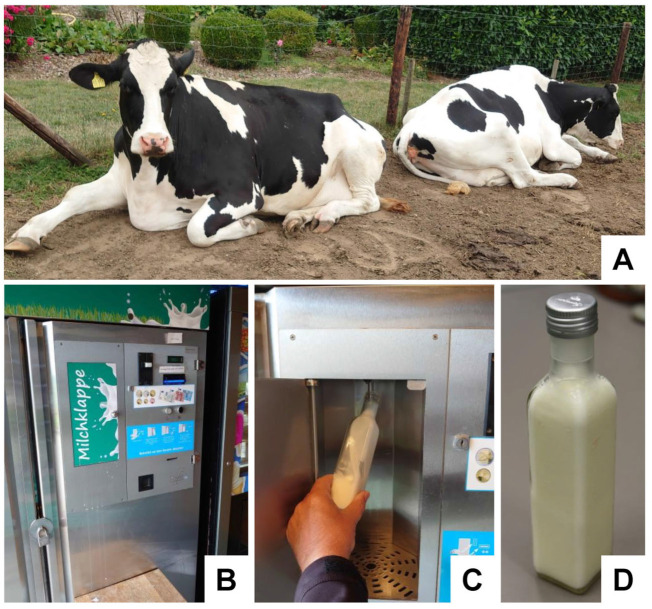
The source of bovine milk as a starting material. (**A**) Milk was taken from German Black Pied cattle (Holstein Schwarzbunt). This race is an endangered dual-purpose cattle breed with a high milk production rate and milk fat content that originates in the North Sea coasts regions of Northern Germany and the Netherlands. (**B**) Appearance of a typical milk vending machine, in which the milk buyer can throw coins into the slot and draw the desired quantity of fresh milk. (**C**,**D**) For our study, we routinely drew 200 mL milk from the respective machines and stored it in standard glass bottles at 4 °C.

**Figure 3 biomedicines-11-02715-f003:**
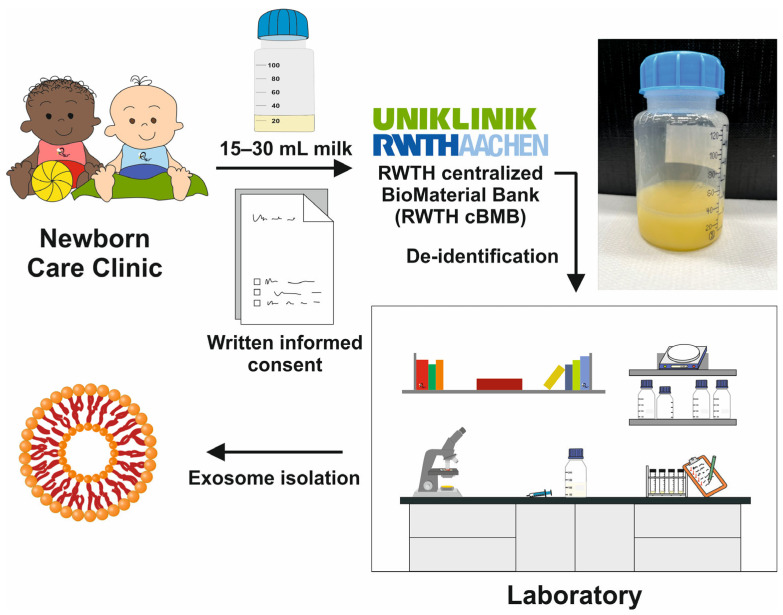
Human breast milk recruitment. The human milk samples (~15–30 mL) were obtained from healthy lactating mothers enrolled in the newborn care clinic of the RWTH University Hospital Aachen. The donated milk samples together with written informed consent forms that allowed the use of the respective samples in our study were transferred to a centralized biomaterial bank. After de-identification, the samples were handled to our laboratory for exosome isolation. The typical appearance of a human milk sample is depicted.

**Figure 4 biomedicines-11-02715-f004:**
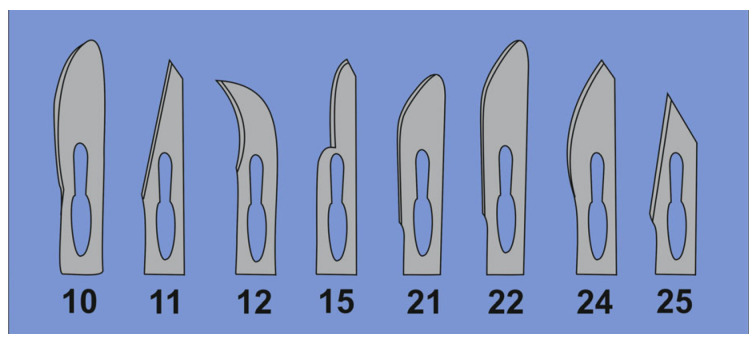
The shapes of the scalpel blades. There is a large variety of scalpel blades differing in length and shape. They carry a shorthand number code that indicates the characteristics of the blade. Scalpels with large curved cutting edge (e.g., blades number 10, 21, and 22) have the highest stability and are recommended.

**Figure 12 biomedicines-11-02715-f012:**
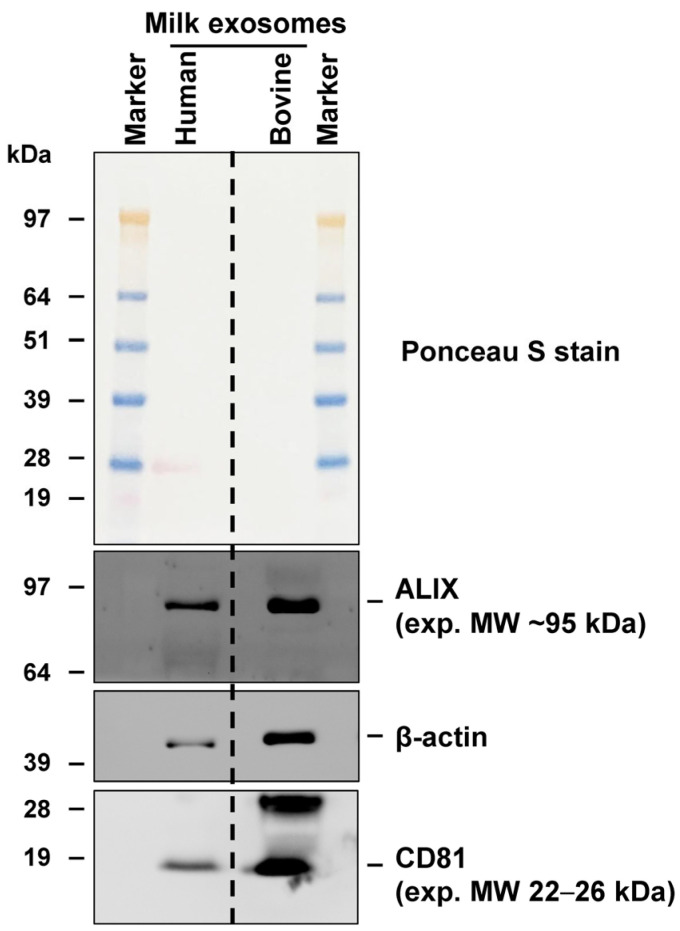
Analysis of typical exosome markers via Western blot analysis. Exosomes were prepared from bovine and human milk according to the outlined protocol. The final exosome fractions (80 µg lysate, as determined with DC protein assay (Bio-Rad Laboratories GmbH, Düsseldorf, Germany; #5000116) were subjected to SDS polyacrylamide gel electrophoresis and tested via Western blot analysis for expression of exosome markers CD81 and ALIX. Ponceau S stain served to document proper protein transfer from the polyacrylamide gel to the 0.2 µm nitrocellulose membrane (Merck, Darmstadt, Germany; #GE10600001). Separation of proteins was achieved with 4–12% Bis-Tris gel using 3-*N*-(morpholino)propanesulfonic acid (MOPS) as running buffer and pre-stained SeeBlue Plus2 protein ladder (ThermoFisher Scientific Inc., Waltham, MA, USA) as size marker.

**Figure 13 biomedicines-11-02715-f013:**
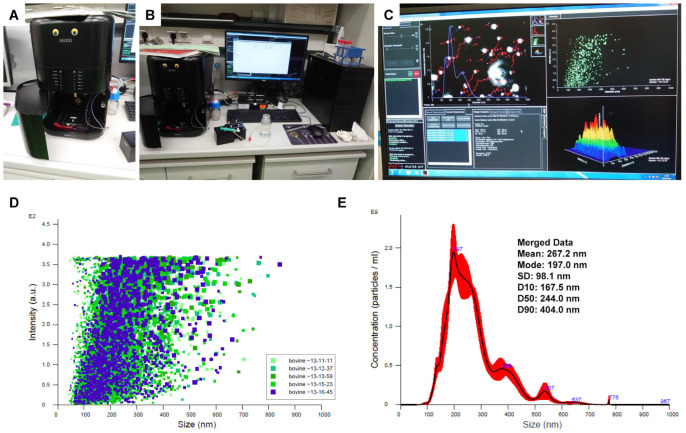
The characterization of exosomes isolated from bovine milk. The exosomes were isolated from bovine milk and the particle size distribution and calculated concentrations were determined via nanoparticle tracking analysis. (**A**) The NanoSight NS300 instrument allows the quick quality characterization of the size (from 10 nm to 1 µm) and concentration (10^6^–10^8^ particles/mL) of extracellular vesicles. (**B**) A typical workspace for nanoparticle tracking analysis (NTA) is shown. (**C**) Screen display during NTA measurement. (**D**) The distribution of particle size in five exosome samples showing that the intensity of the signals is highest in the range of 80–320 nm. (**E**) A representative NTA particle size distribution plot for bovine milk exosomes is shown. The particle size is given in nm (X axis), and the particle concentration is given in ml^−1^ × 10^9^ (Y axis). In the example, the mean size of the exosomes is 267.2 nm ± 98.1 nm. The black line represents the average value of five measurements, while the red areas indicates the standard deviation of the particle size distribution (*n* = 5).

**Figure 14 biomedicines-11-02715-f014:**
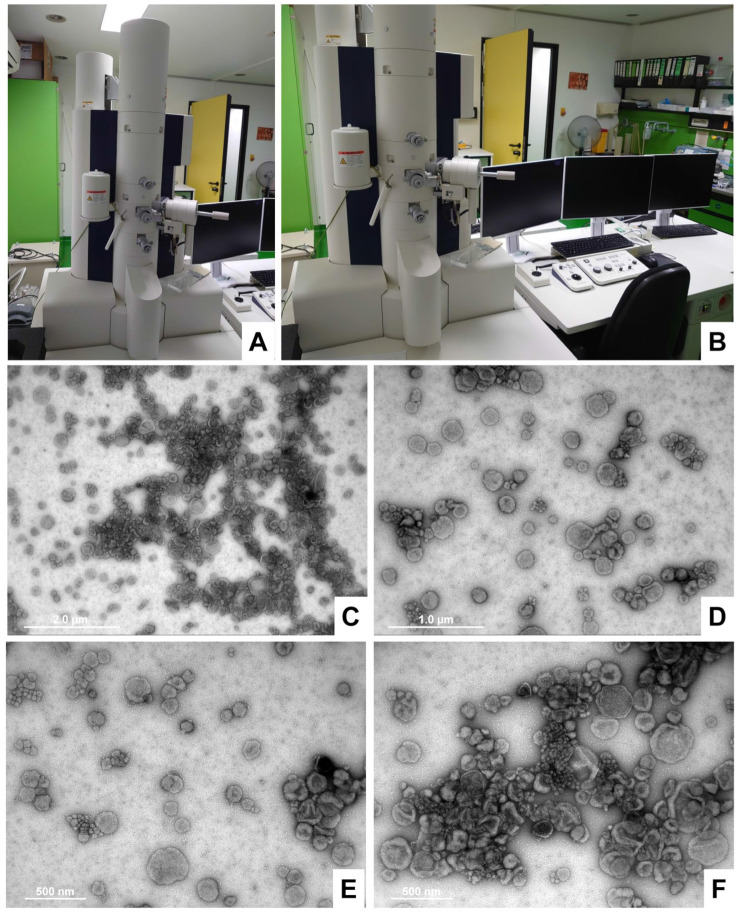
The characterization of exosomes isolated from bovine and human milk via transmission electron microscopy (TEM). (**A**,**B**) Typical electron microscopy equipment and setup used for exosome analysis. In our laboratory, we routinely use the Hitachi HT7800 model. (**C**–**F**) Representative TEM images of isolated exosomes negatively stained with 0.5% uranyl acetate are depicted. Size markers are depicted in each images that were taken at original magnifications of (**C**) 10,000×, (**D**) 20,000×, and (**E**,**F**) 25,000×, respectively.

**Figure 15 biomedicines-11-02715-f015:**
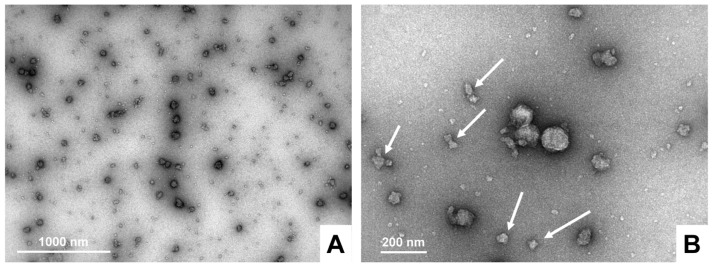
Casein-contaminated bovine milk exosomes. (**A**,**B**) The bovine milk exosomes were purified according to the described protocol with omitting the EDTA step. The exosomes were analyzed via transmission electron microscopy. Contaminating casein aggregates are marked with white arrows. These images were taken at (**A**) 20,000× and (**B**) 50,000×.

## Data Availability

All new experimental data are presented in this article. Additional datasets are available on request from the corresponding author.

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
