# Peer review of "Isolation of Bovine and Human Milk Extracellular Vesicles"

_biomedicines, 2023, doi:10.3390/biomedicines11102715_

Round 1

Reviewer 1 Report

This manuscript presents a protocol for the isolation of exosomes from milk (both bovine and human), although it is not always clear what is novel in the protocol and some efforts should be made to clarify this.

Results pertaining to both bovine and human milk should be presented (only bovine results are presented in both size and concentration it seems)

Extensive use of active present tense is made throughout the manuscript (expeciallly in section  3 where passive tense should be preferred)

Section 5 could be incorporated into the introduction with references, in particular to alleviate the ambiguity over the "large quantities" line 37

Lines 52-69 It is not clear that this paragraph is concerned with milk exosomes only, although casein contamination mentioned line 68 is only a concern in this matrix.

Line 85 Do the authors mean that these methods are not cost effective?

Figure 1. Please be specific on the novelty of the presented flow diageram compared to already published protocols. this is not clear and should be made more central.

Line 158: How relevant is HPES melting point to its lack of stability at high temperature in water?

Section 4.5. What is novel here? Are these methods already published and if so should they be referenced?

Section 4.5.2 Please specify results of size and concentration for human milk. Also, how were these numbers affected when the EDTA step was omitted? As this step seems to be the only novel one in the protocol presented, this information could be important.

Please ensure passive tense is used throughout. For example, section 3.3.2 line 191 could be rewritten as "In this study, tubes were discarded after first use"

Author Response

Dear Reviewer,

many thanks for reviewing our paper in such a short time. We are really grateful for your comments. Please find our response to your comments/suggestions in the attached pdf-file.

Regards

Ralf Weiskirchen

Reviewer 2 Report

The authors have detailed a robust protocol for the extraction of exosomes from both bovine and human milk samples. This procedure relies on a series of meticulously orchestrated standard and ultracentrifugation steps, followed by filtration through 0.2 µm filters and the elimination of any potential contaminants, such as casein aggregates, through treatment with ethylenediaminetetraacetic acid (EDTA). Subsequent analyses, including electron microscopy and nanoparticle tracking, unequivocally affirm the protocol's effectiveness in yielding highly enriched exosome fractions. Furthermore, Western blot analysis has unequivocally confirmed the presence of classic exosomal protein markers, such as CD81 and ALIX, in the purified particles.

The manuscript showcases not only the precision of the experimental work but also the clarity of presentation, with figures that effectively illustrate the findings.

Given the quality of the manuscript, I believe this protocol is suitable for acceptance in its current form.

Author Response

Dear Reviewer 2,

many thanks for your overall favorable comments. Please find our response to your comments in the attached pdf-file.

Many thanks

Ralf Weiskirchen
